# Role of Surgery in Patients with Stage IE Primary Thyroid MALT Lymphoma Staged by a Modified Classification System: The Tokyo Classification

**DOI:** 10.3390/cancers15051451

**Published:** 2023-02-24

**Authors:** Yoshiyuki Saito, Natsuko Watanabe, Nami Suzuki, Naoko Saito, Hiroto Narimatsu, Hiroshi Takami, Kaori Kameyama, Kana Yoshioka, Chie Masaki, Junko Akaishi, Kiyomi Yamada Hames, Masako Matsumoto, Miho Fukushita, Ai Yoshihara, Ritsuko Okamura, Chisato Tomoda, Akifumi Suzuki, Kenichi Matsuzu, Wataru Kitagawa, Mitsuji Nagahama, Jaeduk Yoshimura Noh, Kiminori Sugino, Koichi Ito

**Affiliations:** 1Department of Surgery, Ito Hospital, Tokyo 150-8308, Japan; 2Department of Internal Medicine, Ito Hospital, Tokyo 150-8308, Japan; 3Department of Radiology, Juntendo University, Tokyo 113-8431, Japan; 4Cancer Prevention and Control Division, Kanagawa Cancer Center, Yokohama 241-8515, Japan; 5Graduate School of Health Innovation, Kanagawa University of Human Services, Kawasaki 238-8522, Japan; 6Department of Pathology, Showa University Northern Yokohama Hospital, Yokohama 224-8503, Japan

**Keywords:** mucosa-associated lymphoid tissue lymphoma, primary thyroid lymphoma, involved-site radiation therapy, thyroidectomy

## Abstract

**Simple Summary:**

The therapeutic strategy for and the staging system of primary thyroid mucosa-associated lymphoid tissue (MALT) lymphoma are not established. We conducted a retrospective analysis to (i) establish the appropriate staging system and (ii) assess the role of curative thyroidectomy vs. involved-site radiation therapy (ISRT) after open biopsy in stage IE MALT lymphoma. The modified staging system allows us to distinguish between stages IE and IIE primary thyroid MALT lymphoma. Information about the side effects under ISRT may have limited informative value, but approximately one in three to four patients had radiation-induced permanent complications (mainly dry mouth). Curative thyroidectomy provides good prognoses, equivalent to that of ISRT after open biopsy; it avoids permanent dry mouth, shortens painful periods during treatment, and simplifies ultrasound follow-ups. In the NCCN guidelines, ISRT is the preferred initial therapy for limited-stage non-gastric MALT lymphoma. However, curative thyroidectomy alone can serve as the initial treatment equivalent to ISRT post-open-biopsy.

**Abstract:**

Purposes: To establish the appropriate staging system and assess the role of curative thyroidectomy alone (Surgery) vs. involved-site radiation therapy after open biopsy (OB-ISRT) in stage IE mucosa-associated lymphoid tissue (MALT) lymphoma. Methods: We examined the Tokyo Classification as a modified classification. This retrospective cohort study included 256 patients with thyroid MALT lymphoma; 137 underwent standard therapy (i.e., OB-ISRT) and were enrolled for the Tokyo classification. Sixty stage IE patients with the same diagnosis were examined to compare Surgery with OB-ISRT. Results: Overall survival (*p* = 0.0092) and relapse-free survival (0.00113) were significantly better in stage IE vs. stage IIE under the Tokyo classification. No OB-ISRT and Surgery patients died, but three OB-ISRT patients relapsed. The incidence of permanent complications was 28% in OB-ISRT (mainly dry mouth) and 0% in Surgery (*p* = 0.027). The number of painkiller prescription days was significantly greater in OB-ISRT (*p* < 0.001). During follow-up, the rate of the new appearance/change of the low-density area in the thyroid gland was significantly higher in OB-ISRT (*p* = 0.031). Conclusions: The Tokyo classification allows an appropriate discrimination between stages IE and IIE MALT lymphoma. Surgery can provide a good prognosis in stage IE cases; it also avoids complications, shortens painful periods during treatment, and simplifies ultrasound follow-up.

## 1. Introduction

Primary thyroid lymphoma (PTL) accounts for only 1–2% of extranodal lymphoma and <5% of thyroid malignancies [1,2], and it characteristically occurs in the setting of Hashimoto’s disease [3]. PTL is almost exclusively non-Hodgkin’s, B-cell lymphoma. The most common subtype of PTL is diffuse large B-cell lymphoma (DLBCL) (50–70% of PTLs), followed by mucosa-associated lymphoid tissue (MALT) lymphoma (10–50%) [4].

The therapeutic strategy of primary thyroid MALT lymphoma is not well-established, in part because of primary thyroid MALT lymphoma’s rarity. Treating PTL thus usually follows non-organ-specific guidelines, such as the U.S. National Comprehensive Cancer Network (NCCN) clinical practice guidelines for B-cell lymphomas [5], in which the standard and preferred initial therapy for limited-stage non-gastric MALT lymphoma is involved-site radiation therapy (ISRT) after an open biopsy (OB-ISRT). The guidelines note that surgery may be considered for certain sites including the thyroid.

No staging system for primary thyroid MALT lymphoma exists. The Ann Arbor classification system was initially developed for staging Hodgkin’s disease but has some use in non-Hodgkin lymphoma [6]. After the advent of computed tomography (CT), CT was included in lymphoma staging at the 1989 Cotswolds meeting [7]. Fluorodeoxyglucose (FDG) positron emission tomography (PET)/CT technology is now available in many countries, and in the PET/CT era, the staging for non-Hodgkin lymphoma is the Lugano classification [8], which is based on the Ann Arbor classification. These staging systems have been used clinically for primary thyroid MALT lymphoma, but they cannot distinguish between stage IE and stage IIE appropriately in terms of prognosis, as we have de-scribed [4,9].

Ito Hospital (Tokyo) specializes in the treatment of thyroid diseases and has long collected clinical data on PTL cases [4,9,10,11]. The objectives of the present study were to (i) establish the appropriate staging system for primary thyroid MALT lymphoma, and (ii) assess the role of curative thyroidectomy in patients with stage IE primary thyroid MALT lymphoma compared to OB-ISRT.

## 2. Materials and Methods

### 2.1. Assessment of a Modified Staging System for Limited-Stage Thyroid MALT Lymphoma

#### Study Design and Patients

We conducted a retrospective cohort study of patients with limited-stage primary thyroid MALT lymphoma at Ito Hospital (Tokyo) between January 1990 and October 2021 after obtaining Institutional Ethics Board approval (no. 169). Patients with limited-stage MALT lymphoma were included for the modified staging system’s assessment. The preferred initial therapy for limited-stage non-gastric MALT lymphoma is OB-ISRT [5]. Our exclusion criteria were thus patients who had undergone (1) a thyroidectomy instead of an open biopsy for diagnosis or (2) initial treatment(s) other than OB-ISRT. The pathological diagnoses were based on the World Health Organization classification of lymphoid neoplasms.

### 2.2. The Conventional and Novel Staging Systems and the Data Collection

Stage IE primary thyroid MALT lymphoma is defined as involvement of the thyroid alone; stage IIE is defined as the major site of involvement being in the thyroid gland with associated regional lymph node involvement (cervical and superior mediastinal areas). To divide limited-stage MALT lymphoma into clinical stage IE and stage IIE, the conventional staging systems, i.e., the Lugano classification [8] and the Ann Arbor classification system with Cotswold modifications [7], recommend using CT and/or PET/CT. The Lugano classification states that unexplained node enlargement detected by a CT scan and/or increased FDG uptake shown by PET/CT in the regional lymph nodes indicates clinical stage IIE; however, the classification does not mention what size of lymph node is considered an unexplained enlarged lymph node. The conventional staging systems thus involve qualitative and subjective judgments.

Moreover, the coexistence of Hashimoto’s disease makes it difficult to distinguish ‘involved’ from ‘inflamed’ lymph nodes. A minimal axial diameter of 10 mm is generally thought to be the most accurate size criterion for predicting lymph node metastasis [12]. After we held a discussion in Tokyo among leading endocrinologists, a hematologist, endocrine surgeons, a radiologist, and a pathologist regarding the appropriate objective staging system, we defined ‘clinical involvement of the regional lymph nodes of the thyroid’ as follows as the “Tokyo classification”: (1) a minimal axial dia. ≥10 mm on CT, and/or (2) increased uptake on 18F-FDG PET and/or 67Ga-citrate scintigraphy (Table 1). For a comparison with this new definition, we collected data on the patients’ stages from their medical records in our hospital’s PTL database; those stages were based on one of the two above-described conventional staging systems.

## 3. Comparison of OB-ISRT and Surgery

### 3.1. Study Design and Patients

We next compared surgery (i.e., curative thyroidectomy) alone with OB-ISRT in the stage IE patients defined using the Tokyo classification. Because a thyroidectomy without regional lymph node dissection is not curative for stage IIE, we did not include stage IIE patients in this analysis. Initially, at Ito Hospital, all patients who underwent a thyroidectomy for limited-stage primary thyroid MALT lymphoma also underwent post-operative radiotherapy. After 2012, we provided a therapeutic option: no ISRT after a curative thyroidectomy for stage IE MALT lymphoma.

Given this background, the inclusion criteria were patients with stage IE MALT lymphoma who had undergone OB-ISRT or a thyroidectomy alone during the period between January 2012 and October 2021. The exclusion criteria were: (1) patients whose data concerning treatment complications could not be collected (e.g., lack of radiation therapy complication information because the patient underwent an open biopsy at Ito Hospital but ISRT at another institution), and (2) patients who had another disease associated with lymphadenopathy (other than autoimmune thyroid disease [Graves’ disease and Hashimoto’s disease]).

### 3.2. Study Variables

The patient demographics included age, sex, the coexistence of Hashimoto’s disease, hypothyroidism before treatment, and lesion location. We also analyzed variables pertaining to treatment-related complications, the new appearance or change of a low-density area (LDA) on the ultrasound post-treatment, the performance of a biopsy (Bx)/fine-needle aspiration (FNA) or scintigraphy due to the suspicion of recurrence, and cancers after treatment. These data were collected retrospectively from the medical records. Radiation-induced complications were graded using CTCAE ver. 5.0 [13].

Most of the patients with ISRT underwent radiotherapy with 30 Gy at Ito Hospital, but some patients underwent ISRT at other institutions nearby. Figure 1 shows the typical radiation treatment planning for stage IE primary thyroid MALT lymphoma at Ito Hospital.

### 3.3. Statistical Analyses

We used the χ^2^-test or Fisher’s exact test to compare nominal data, and we used the Kaplan–Meier method and stratified log-rank tests for the overall survival (OS) and relapse-free survival (RFS) rates. For all procedures, *p*-values < 0.05 were considered significant. The statistical analyses were conducted using STATA software ver. 15.0 (Stata, College Station, TX, USA).

## 4. Results

### 4.1. The Modified Staging System for Limited-Stage Thyroid MALT Lymphoma

The selection process is summarized in Appendix A. Between January 1990 and October 2021, 256 patients were diagnosed with limited-stage MALT lymphoma at Ito Hospital, and 119 were excluded from the study based on the exclusion criteria: thyroidectomy performed for diagnosis (*n* = 59), and 60 patients underwent initial treatments other than ISRT after open biopsy (combined modality treatments, *n* = 50; observations, *n* = 4; chemoimmunotherapies, *n* = 4; and rituximab, *n* = 2). Thus, 137 patients were included in the analyses. Appendix A summarizes these patients’ baseline characteristics: median age, 71 years; 85.4% were female; 87.6% had Hashimoto’s disease; and 2.9% had Grave’s disease.

Figure 2 depicts the patients’ OS and RFS rates, comparing the rates of the patients with stage IE and stage IIE disease classified with one of the conventional staging systems or the Tokyo classification. When the staging was done with the conventional staging systems, there was no significant difference in the OS of the stages IE and IIE patients (hazard ratio [HR] 3.05, 95% confidence interval [CI]: 0.87–10.69; *p* = 0.0673), although the OS in stage IIE tended to be worse compared to that in stage IE. The RFS was also not significantly different between stages IE and IIE under the conventional staging systems (HR 1.76, 95%CI: 0.69–4.53; *p* = 0.232).

After the restaging using the Tokyo classification, the 5- and 10-year OS rates were 96.8% and 95.0% for stage IE, and 78.1% and 78.1% for stage IIE, respectively (HR 4.29, 95%CI: 1.30–14.14; *p* = 0.0092). The 5- and 10-year RFS rates were 94.7% and 89.7% for stage IE and 74.1% and 69.8% for stage IIE, respectively (HR 3.18, 95%CI: 1.24–8.17; *p* = 0.00113) under the Tokyo classification. After the restaging using the Tokyo classification, no upstaging was detected, but downstaging from stage IIE to stage IE was observed in 17 patients (33.3%). No relapse was detected in these 17 cases with a median follow-up of 8.1 years.

### 4.2. OB-ISRT vs. Surgery

To assess the advantages/disadvantages of curative thyroidectomy for primary thyroid MALT lymphoma, we next compared surgery alone to OB-ISRT in the stage IE patients. Appendix A summarizes the selection process. Under the Tokyo classification, 73 stage IE patients underwent either OB-ISRT or surgery alone between January 2012 and October 2021. We excluded 13 of these patients based on the exclusion criteria and compared the final totals in the Surgery (*n* = 14) and OB-ISRT (*n* = 46) groups. Appendix A provides the patients’ baseline characteristics. There were no significant between-group differences in age, sex, or the coexistence of Hashimoto’s disease.

Both radiation therapy and thyroidectomy have treatment-specific complications. For radiation therapy to the cervical area, well-known radiation-induced complications are dermatitis, oral mucositis, dry mouth, dysphagia, esophagitis/pharyngitis, and hoarseness. Possible surgical complications after thyroidectomy are hypoparathyroidism and recurrent laryngeal nerve paralysis (RLNP). Both treatments also pose a risk of thyroid dysfunction. Table 2 and Appendix A summarizes the treatment-related complications that occurred in the Surgery and OB-ISRT groups. Regarding the complications other than thyroid function, all 46 patients (100%) in the OB-ISRT group suffered from a treatment-related complication, whereas surgical complications occurred in 36% of the Surgery group (p < 0.001). Radiation-induced esophagitis/pharyngitis (98% of the patients), dysphagia (85%), dermatitis (54%), dry mouth (50%), hoarseness (33%), and oral mucositis (11%) were observed.

Almost all of the cases of dermatitis and/or hoarseness were CTCAE grade 1 and did not require any specific treatment, but many of the cases of esophagitis/pharyngitis, dysphagia, dry mouth, and/or oral mucositis were CTCAE grade 2 and needed additional treatment. Most of the radiation-induced complications improved with time, but 28% of patients with OB-ISRT had permanent complications, most of which were dry mouth. One patient in the OB-ISRT group had post-operative bleeding and required re-operation after the open biopsy.

In the Surgery group, all cases of surgical complications were transient hypoparathyroidism (incidence rate: 36%), and no patients faced permanent hypoparathyroidism or post-operative RLNP. The incidence of permanent complications other than thyroid function was thus significantly higher in the OB-ISRT group compared to the Surgery group (*p* = 0.027). The number of painkiller prescription days was significantly greater in the OB-ISRT group vs. the Surgery group (*p* < 0.001).

Concerning complications associated with thyroid function, only 15% and 21% of the patients in the OB-ISRT and Surgery groups, respectively, were euthyroid after treatment (*p* = 0.685). There were also no significant differences between the groups in terms of the incidence of new hypothyroidism or in the rate of patients whose levothyroxine dose was increased after treatment.

The patients’ post-treatment courses are described in Table 3. During the follow-up, the rate of a new appearance or change of an LDA in the thyroid gland after OB-ISRT was significantly higher than that after surgery alone (48% vs. 14%; *p* = 0.031). However, there were no significant differences between the OB-ISRT and Surgery groups in the post-treatment performance of Bx/FNA (15% vs. 7%) or scintigraphy (11% vs. 0%, respectively). After the initial treatment for lymphoma, three patients in the OB-ISRT group had cancers (lung, gastric, and colorectal cancers, respectively). During the follow-up (93.3% follow-up rate with an average follow-up of 5.12 years), none of the patients died, and three OB-ISRT patients had a relapse of MALT lymphoma (two in the thyroid gland and one in a lymph node) (Figure 3).

## 5. Discussion

The results of our analyses demonstrate that the modified staging system (i.e., the Tokyo classification) is suitable for primary thyroid MALT lymphoma compared to the conventional staging systems (Ann Arbor and Lugano), which are non-organ-specific systems. Under the Tokyo classification, both curative thyroidectomy alone and OB-ISRT provided a satisfactory prognosis.

Because the morphological and immunohistological findings are sometimes insufficient for a confident diagnosis in terms of regional lymph nodes, the pathological diagnosis of lymph node involvement in MALT lymphoma is more difficult than that of lymph node involvement in DLBCL. Moreover, FDG PET/CT has limited sensitivity in the assessment of MALT lymphoma because of the low metabolic rate of this malignancy and the high rate of concurrent Hashimoto’s disease [14]. For these reasons, in patients with primary thyroid MALT lymphoma, it is difficult to determine the benign vs. malignant status of lymph nodes precisely by imaging examinations or by pathological surveys. There is thus a possibility that some of the present patients classified as having stage IE disease under our modified staging system were not truly stage IE (i.e., had lymph node involvement). However, this issue is not a concern in clinical settings, because patients who undergo a curative thyroidectomy without additional therapy for small (<10 mm) lymph nodes have good prognoses. This may be because MALT lymphoma is an indolent and low-grade malignancy. Using the Tokyo classification, 33.3% of the stage IIE cases in this study were restaged as stage IE, and it would thus be possible to reduce the irradiation field size for these patients.

There is currently no consensus as to which treatment is the best for stage IE primary thyroid MALT lymphoma. Our present analyses of stage IE patients with primary thyroid MALT lymphoma revealed that both curative thyroidectomy alone and OB-ISRT provided satisfactory overall and relapse-free survival. It is thus reasonable for physicians to present both treatments as therapeutic options to these patients.

Because ISRT and thyroidectomy have different types of treatment-related complications, it is difficult to state which treatment is better in terms of complications. Nonetheless, our present findings demonstrate that thyroidectomy alone was superior in terms of permanent complications and pain duration compared to OB-ISRT. In this series, radiation-induced permanent complications (mainly dry mouth) occurred in approx. one in three to four patients. Most patients who have primary thyroid MALT lymphoma are in the seventh and eighth decades of life [15], and dry mouth is known to cause an impaired quality of life among older people [16,17]. Our finding that the patients who underwent OB-ISRT used painkillers for a longer period seems to have been caused by the longer-term treatment of fractionated radiation compared to surgery. Regarding thyroid function after treatment, the present results revealed that neither surgery nor OB-ISRT is particularly better than the other treatment.

Although an appropriate radiation dose for primary thyroid MALT lymphoma has not been established, the NCCN guidelines recommend 24–30 Gy for MALT lymphoma [5]. In a second analysis of the present patient series, the median radiation dose was approx. 30 Gy (Appendix A). A reduction of the radiation dose (e.g., to 24 Gy) could thus be considered toward the goal of reducing radiation-associated complications. However, an increase in the relapse rate within the thyroid gland due to a radiation dose reduction may be possible, as a few patients in the present OB-ISRT group had a relapse. As noted in a textbook, the tolerance doses for 5% and 50% xerostomia in 5 years are 32 and 46 Gy, respectively [18]. In the present patient series, though the median radiation dose was ~30 Gy and the irradiation to parotids was minimized in most cases (as shown in Figure 1), the incidence of dry mouth was higher than expected.

If a patient’s clinical features [19] and/or the results of an FNA [20] suggest a lesser possibility of DLBCL, an open biopsy and thyroidectomy tend to be preferred compared to a core-needle biopsy, to ensure that aggressive histologies are not missed since the more limited core-biopsy specimen may not be representative of the entire tumor [14]. These patients thus need to undergo surgery (i.e., open biopsy or thyroidectomy) at least one time. Compared to a thyroidectomy, an open biopsy cannot avoid blind manipulation. For some surgeons, this causes anxiety about injuring organs behind the thyroid such as the trachea. In our present series, a serious surgical complication (Clavien–Dindo classification [21] ≥grade III) occurred not in the Surgery group but in the OB-ISRT group. An open biopsy is thus not always a less-invasive and safer surgical procedure compared to thyroidectomy.

The potential advantages of a curative thyroidectomy are (i) easy ultrasound follow-up due to the removal of LDA(s) in the thyroid gland, (ii) a more definite removal of lymphoma within the thyroid gland, and (iii) the removal of the risk factor of lymphoma when a total thyroidectomy is performed. After ISRT, an LDA inside the thyroid gland does not disappear soon and remains in some cases. In such cases, careful observation of the lesions is required. For this reason, the removal of LDA(s) by a thyroidectomy provides the possibility of relieving both patients’ and physicians’ psychological stress caused by LDAs remaining after treatment. Moreover, a curative thyroidectomy may help rule out histological transformation. Rajamäki et al. reported that a rebiopsy based on a high SUVmax in FDG-PET/CT was valuable in detecting the transformation of follicular lymphoma [22], but it is not yet clear whether FDG-PET/CT can detect the transformation of a primary thyroid MALT lymphoma. Helicobacter pylori plays a major role in the pathogenesis of gastric MALT lymphoma, and antibiotic therapy for H. pylori is recommended for H. pylori-positive gastric MALT lymphoma [5]. By the same principle, the removal of thyroiditis, i.e., a total thyroidectomy, might have potential merit for preventing a relapse inside the thyroid gland because primary thyroid MALT lymphoma characteristically occurs in the setting of Hashimoto’s disease. Compared to total thyroidectomy, the advantages of hemithyroidectomy are (i) the avoidance of post-operative hypoparathyroidism and bilateral vocal cord paralysis, and (ii) the possibility of maintaining euthyroid status after surgery.

### Study Limitations

Despite our encouraging results, some study limitations should be noted: (i) We did not assess whether curative thyroidectomy following an open biopsy is an acceptable treatment. (ii) The data on radiation-induced complications were collected from the patients’ medical records and not questionnaires completed by the patients. (iii) We did not evaluate the appropriate radiation dose for each stage of primary thyroid MALT lymphoma.

All of the patients in the Surgery group underwent one-stage surgery, i.e., a thyroidectomy, for diagnosis and treatment, and none of the patients underwent a thyroidectomy alone after a diagnosis by open biopsy. However, we feel that performing a thyroidectomy after a diagnosis by open biopsy is also an acceptable therapeutic option. In addition, an open biopsy exposes only the front side of the thyroid and does not expose the recurrent laryngeal nerve (RLN); a subsequent thyroidectomy can thus be performed without adhesion around the RLN [23]. We thus do not think that a thyroidectomy after an open biopsy increases the incidence of complications compared to the one-stage surgery.

This was a retrospective study and did not use any questionnaires completed by the patients to collect the data on radiation-induced complications. Hospital records usually underestimate the true number of complications. As noted above, we could not assess the appropriate radiation dose with a good balance of reducing radiation-associated complications and avoiding relapse within the thyroid gland because few patients in this series received 24 Gy radiotherapy. Because ISRT is the preferred initial therapy in the NCCN guidelines, the present Surgery group was quite smaller than the OB-ISRT group (14 vs. 46 patients, respectively).

## 6. Conclusions

Our study results demonstrated that the modified staging system (the Tokyo classification) allowed us to distinguish between stage IE and stage IIE. Although our findings regarding ISRT side effects may have limited informative value due to this study’s retrospective nature, approx. one in three to four patients had radiation-induced permanent complications (mainly dry mouth) in this series. The use of the Tokyo classification revealed that (i) curative thyroidectomy provides a good prognosis that is equivalent to that provided by OB-ISRT, and it also avoids permanent dry mouth, shortens painful periods during treatment, and simplifies the ultrasound follow-up in stage IE patients; and (ii) approx. three in 10 stage IIE patients were reclassified as stage IE and would thus be able to choose thyroidectomy alone as a therapeutic option, although, in the NCCN guidelines, ISRT is the preferred initial therapy for limited-stage non-gastric MALT lymphoma. Further research is necessary to determine the precise potential implications of both curative thyroidectomy and the clinical usage of the Tokyo classification.

## Figures and Tables

**Figure 1 cancers-15-01451-f001:**
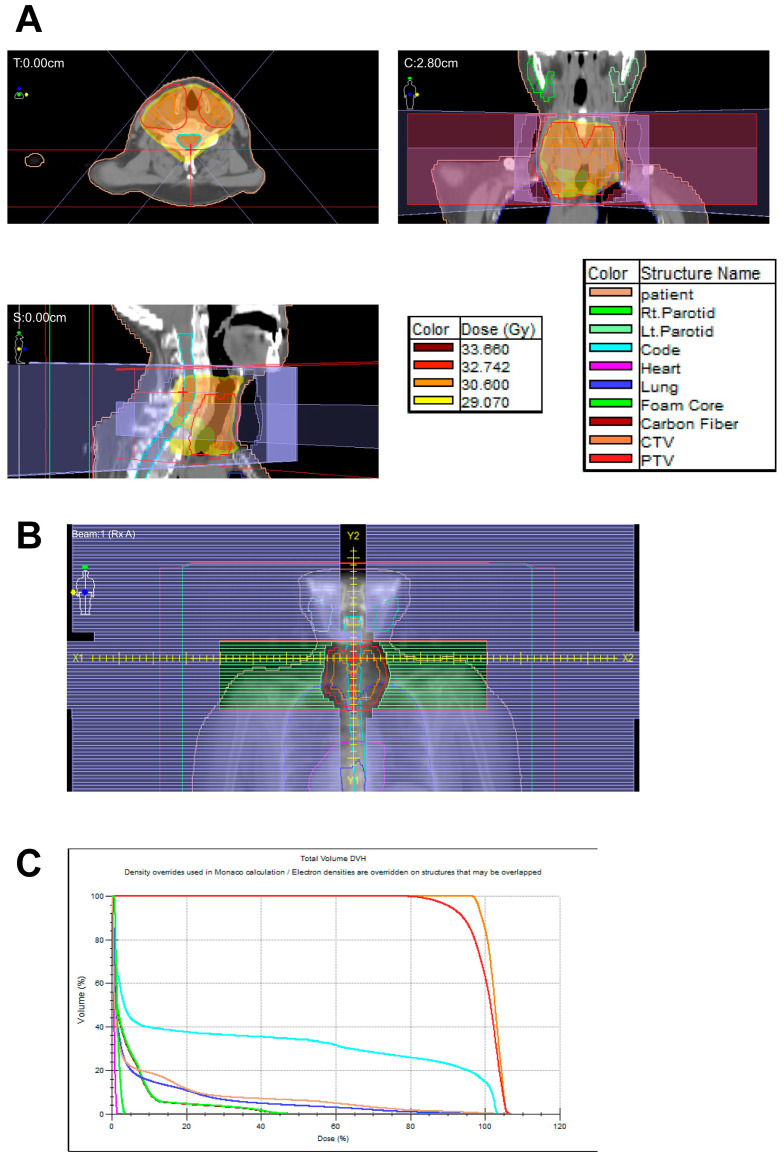
Typical radiation treatment planning for stage IE primary thyroid mucosa-associated lymphoid tissue (MALT) lymphoma at Ito Hospital. Tables of color correspondence (**A**) Radiation dose distribution on the treatment planning CT image. The lower right is the corresponding table between color lines in (**A**–**C**) and structure names. Orange line: the clinical target volume (CTV). Red line: the planning target volume (PTV). (**B**) Beam’s eye view demonstrates that the radiation protocol for stage IE minimizes salivary gland irradiation outside of the PTV. (**C**) The cumulative dose-volume histogram (DHV) also shows that the radiation doses toward the right and left parotids (yellowish-green curves) are minimized.

**Figure 2 cancers-15-01451-f002:**
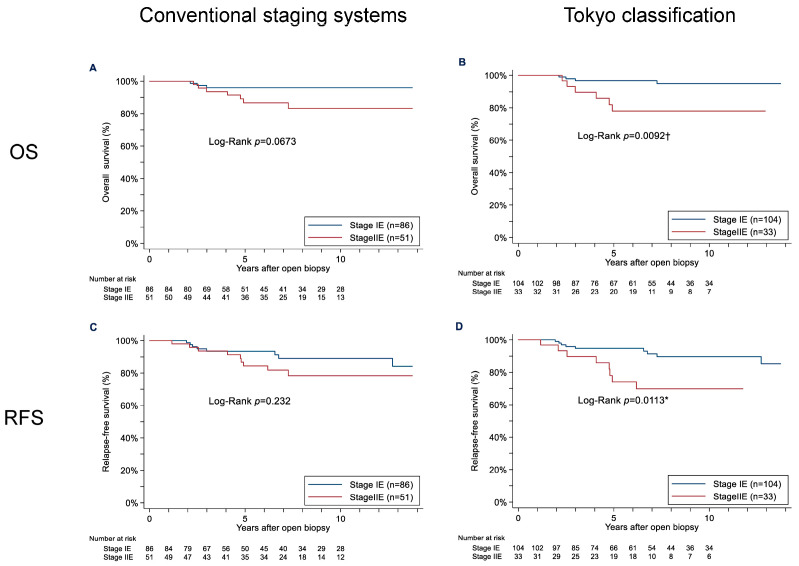
The overall survival (OS) and relapse-free survival (RFS) curves for the patients with stage IE and stage IIE primary thyroid mucosa-associated lymphoid tissue (MALT) lymphoma classified with the conventional staging systems (**A**,**B**) and the modified staging system (the Tokyo classification) (**C**,**D**). * *p* < 0.05. † *p* < 0.001.

**Figure 3 cancers-15-01451-f003:**
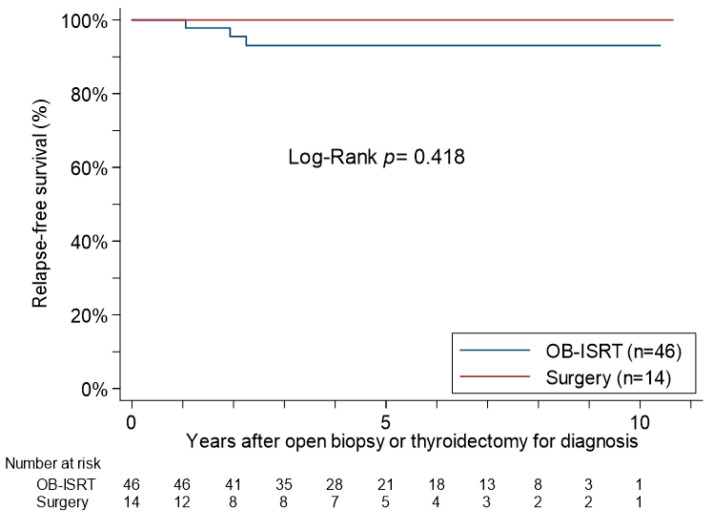
Relapse-free survival curves for the stage IE primary thyroid MALT lymphoma patients who underwent OB-ISRT and those who underwent Surgery after an open biopsy or thyroidectomy.

**Table 1 cancers-15-01451-t001:** Modified staging system for primary thyroid MALT lymphoma: The Tokyo Classification.

Stage	Definition	CT-Based Staging	PET/CT-Based Staging ^a^
IE	Involvement of the thyroid alone	Minimal axial dias. of all regional lymph nodes are <10 mm	No regional lymph node with increased FDG uptake
IIE	Major site of involvement in the thyroid with associated regional lymph node	A minimal axial dia. of a regional lymph node is ≥10 mm	Increased FDG uptake in a regional lymph node

^a^ In the cases in which PET/CT is not available (such as old cases), using 67 Ga-citrate scintigraphy instead is acceptable. CT: computed tomography, FDG: fluorodeoxyglucose, MALT: mucosa-associated lymphoid tissue, PET: positron emission tomography.

**Table 2 cancers-15-01451-t002:** Comparison of treatment-related complications between the OB-ISRT and Surgery groups.

	OB-ISRT *n* = 46	Surgery *n* = 14	*p*
Complications without thyroid function			
Any complications	46 (100)	5 (36)	<0.001 ^†^
Radiation-induced complications, n (%): G1/G2/G3 ^a^			
Dermatitis	25 (54): 24/1/0	–	
Oral mucositis	5 (11): 0/5/0	–	
Dry mouth	23 (50): 5/18/0	–	
Dysphagia	39 (85): 9/30/0	–	
Esophagitis/Pharyngitis	45 (98): 4/41/0	–	
Hoarseness	15 (33): 12/2/1	–	
Complications associated with open biopsy			
Post-operative bleeding	1 (2)		
Surgical complications			
Transient hypoparathyroidism	–	5 (36)	
RLNP	–	0 (0)	
Permanent complications	13 (28)	0 (0)	0.027 *
Prescribed painkillers, days	12 (3, 18)	2 (2, 5)	<0.001 ^†^
Complications associated with thyroid function			
Euthyroid after treatment	7 (15)	3 (21)	0.685
New hypothyroidism	16 (35)	7 (50)	0.305
Increased LT4 dose after treatment	31 (67)	10 (71)	1.000

Data are number (%) or median (IQR). * *p* < 0.05. ^†^
*p* < 0.001. ^a^ CTCAE ver. 5.0 grade 1, 2, and 3. CTCAE: Common Terminology Criteria for Adverse Events, IQR: interquartile range, LT4: levothyroxine, OB-ISRT: involved-site radiation therapy after open biopsy, RLNP: recurrent laryngeal nerve paralysis.

**Table 3 cancers-15-01451-t003:** Comparison of the post-treatment course between the OB-ISRT and Surgery groups.

	OB-ISRT *n* = 46	Surgery *n* = 14	*p*
New appearance or change of low-density area on US	22 (48)	2 (14)	0.031 *
Suspected recurrence	8 (17)	1 (7)	0.671
Performance of Bx or FNA due to suspicion of recurrence	7 (15)	1 (7)	0.667
Performance of scintigraphy due to suspicion of recurrence	5 (11)	0 (0)	0.329
Cancers other than lymphoma after treatment	3 (7)	0 (0)	1.000

Data are number (%). * *p* < 0.05. Bx: biopsy, FNA: fine needle aspiration, OB-ISRT: involved-site radiation therapy after open biopsy, US: ultrasound.

## Data Availability

The study’s data are available upon request from the corresponding author.

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
