# Peer review of "Role of Surgery in Patients with Stage IE Primary Thyroid MALT Lymphoma Staged by a Modified Classification System: The Tokyo Classification"

_cancers, 2023, doi:10.3390/cancers15051451_

Round 1

Reviewer 1 Report

This is a novel manuscript. Comments:

-More details of the radiotherapy dosing should be in the body of the manuscript (in Methods section). The doses used are at the upper end of recommendations and this may have impacted on the toxicity observed. There is some comment on this at the end of the Discussion section but should be earlier in this section.

Author Response

Response to Reviewer 1 Comments

Point 1: More details of the radiotherapy dosing should be in the body of the manuscript (in Methods section).

Response 1: Thank you for your valuable comments. We have added the details with a figure concerning the radiation treatment planning (Pages 3 and 4, and Fig. 1).

Point 2: The doses used are at the upper end of recommendations and this may have impacted on the toxicity observed. There is some comment on this at the end of the Discussion section but should be earlier in this section.

Response 2: The comment has been moved to the earlier in the Discussion section (Page 9).

Reviewer 2 Report

This is well a written paper regarding an importatnt issue in a rare lymphoma subtype. The knowledge of radiotherapy associated late effects is increasing and is more and more regarded as important aspect decreasing patients quality of life. And I recommend acceptance of this manuscript after revisions.

I have one major comment. Authors regard they classification as a new classification. I would regard it merely a precision of an existing classification.

Underneath are some minor cooments:

1) page 3lines 105-106 increased uptake in PET-CT is one of the criterion. This should be further defined, what is considered an increased uptake.

2) Page 3, line 138. Authors should define how the data of complicatios was gathered. Was it collected retrospectively from hospital records or did the patient fill some questionnaires? Usually hospital records underestimate the true number of complications.

3) Table 2. I consider, that radiation therapy associated long term complicatiosn posses a crucial guestions, more important than the reversible acute ones. Long term complicatiosn increase also during time. For this reason I consider authors should report also the lenght of follw-up  and cumulative incidence analyzed by Kapplan-Meier method. Also the type of long term complicatiosn should be presented. I would also point out the three other cancers in radiotherapy group.

4) Page 7 lines 237-239. There is perhaps some extra phrases left from the intructions??

5) In discussion  it would be interesting to discuss more about the benefits/risks of subtotal versus total thyreoidectomy.

6) Authors justify total thyreoidectomy by the possibility to rule out histological transformation. In this context it would be relevant to refer to a  work pointing the potential of PET-CT in ruling out histological transformation.

FDG-PET/CT-guided rebiopsy may find clinically unsuspicious transformation of follicular lymphoma

Rajamäki, A., Kuitunen, H., Sorigue, M., ...Kuittinen, O., Sunela, K. Cancer Medicine, 2023, 12(1), pp. 407–411

Author Response

Response to Reviewer 2 Comments

Point 1: I have one major comment. Authors regard they classification as a new classification. I would regard it merely a precision of an existing classification.

Response 1: We agree and now use ‘modified’ instead of ‘new’ to describe the Tokyo classification.

Point 2: Page 3, lines 105-106. Increased uptake in PET-CT is one of the criterion. This should be further defined, what is considered an increased uptake.

Response 2: According to an earlier study (doi: 10.1007/s12149-012-0669-1), the cut-off delayed SUVmax value of 4.0 might differentiate the lymph nodes between malignant lymphoma and benign lesions. However, the proper cut-off value depends on the scan timing, and the timing of the scan after 18F-FDG injection varies with the PET/CT facilities. We thus did not use this cut-off value but instead used the radiogram interpretation report data of each PET/CT facility. Just in case, we rechecked all of the PET/CT data, and we found that the staging did not change even if the cut-off value of 4.0 was used for all PET/CT (ignoring scan timing) instead of radiogram interpretation report data.

Point 3: Page 3, line 138. Authors should define how the data of complications was gathered. Was it collected retrospectively from hospital records or did the patient fill some questionnaires? Usually hospital records underestimate the true number of complications.

Response 3: Thank you for this important point. This was a retrospective study and did not use any questionnaires. We now mention this in the Methods section and as a study limitation (Page 3 and 10).

Point 4: Table 2. I consider, that radiation therapy associated long term complications posses a crucial questions, more important than the reversible acute ones. Long term complications increase also during time. For this reason I consider authors should report also the length of follow-up and cumulative incidence analyzed by Kaplan-Meier method. Also the type of long term complications should be presented. I would also point out the three other cancers in radiotherapy group.

Response 4: The permanent complications include (1) acute complications that occur within ≥2–3 months and persist permanently, and (2) late complications. In general, patients usually complain about dry mouth around 2–3 weeks after starting radiation therapy, while chronic radiation-induced skin problems can occur >1 year later. It is thus difficult to compare different types of permanent complications using the Kaplan-Meier method. Instead, we have added a new table concerning permanent complications (Suppl. Table S3). We have also added information about the length of follow-up, the three other cancers (Page 7), and the type of long term complications (Suppl. Table S3).

Point 5: Page 7 lines 237-239. There is perhaps some extra phrases left from the instructions??

Response 5: We apologize for the careless error. We have deleted the instructions.

Point 6: In discussion it would be interesting to discuss more about the benefits/risks of subtotal versus total thyroidectomy.

Response 6: No patients in this series underwent a subtotal thyroidectomy. Instead, we have provided more text about the benefits/risks of hemithyroidectomy versus total thyroidectomy (Page 9).

Point 7: Authors justify total thyroidectomy by the possibility to rule out histological transformation. In this context it would be relevant to refer to a work pointing the potential of PET-CT in ruling out histological transformation

FDG-PET/CT-guided rebiopsy may find clinically unsuspicious transformation of follicular lymphoma.

Rajamäki, A., Kuitunen, H., Sorigue, M., ...Kuittinen, O., Sunela, K. Cancer Medicine, 2023, 12(1), pp. 407–411

Response 7: We now mention this and have cited the article (Page 9).

Round 2

Reviewer 1 Report

all questions/comments have been addressed

Author Response

Response to Academic Editor Comments

Point 1: In the summary and conclusion, the authors should add 1. the recommendation for ISRT according to NCCN guidelines and 2. the limited informative value of the side effects under radiotherapy.

Response 1: We have added this information to the simple summary (Page 1) and the Conclusion section (Page 11).

Reviewer 2 Report

I suggest accepting this manuscript in its current version